

# Bedrock incision by bedload: insights from direct numerical simulations

Guilhem Aubert[1], Vincent J. Langlois[1], and Pascal Allemand[1]

[1]Laboratoire de Géologie de Lyon, Université Claude Bernard Lyon 1 / ENS de Lyon / CNRS UMR5276, Villeurbanne, France.

*Correspondence to:* Vincent J. Langlois, Laboratoire de Géologie de Lyon, 2 rue Raphaël Dubois, 69100 Villeurbanne, France. (vincent.langlois@univ-lyon1.fr)

**Abstract.** Bedload sediment transport is one of the main processes that contribute to bedrock incision in a river and is therefore one of the key control parameters in the evolution of mountainous lanscapes. In recent years, many studies have addressed this issue through experimental setups, direct measurements in the field or various analytical models. In this article, we present a new direct

numerical approach: using the classical methods of discrete element simulations applied to granular materials, we compute explicitly the trajectories of a number of pebbles entrained by a turbulent water stream over a rough solid surface. This method allows us to extract quantitatively the amount of energy that successive impacts of pebbles deliver to the bedrock, as a function of both the amount of sediment available and the Shields number. We show that we reproduce qualitatively the behaviour

observed experimentally by Sklar and Dietrich (2001) and observe both a 'tool-effect' and a 'cover-effect'. Converting the energy delivered to the bedrock into an average long-term incision rate of the river leads to predictions consistent with observations in the field. Finally, we reformulate the dependency of this incision rate with Shields number and sediment flux, and predict that the cover term should decay linearly at low sediment supply and exponentially at high sediment supply.

## 1 Introduction

The incision of bedrock channels is one of the key processes that govern the formation and evolution of mountain ranges (Anderson, 1994; Howard, 1994; Whipple and Tucker, 1999). Long-term averaged incision rates can take values from $0.02$ to $14\,\text{mm/yr}$ (see for instance the review by Lague (2014)). It has also been observed that under rarely reached conditions, the short-term incision rate

can reach even higher values, up to a few metres per day (Hartshorn et al., 2002; Lamb and Fonstad, 2010; Cook et al., 2014). In order to model the long-term evolution of the morphology of bedrock



rivers, and more generally of mountainous landscapes, it is often necessary to adopt a simple macroscopic law to take into account the process of bedrock incision. One of the most commonly used approach, the stream-power incision model, assumes that the incision rate within a river channel

varies as a power law of both its local slope and its drainage area (which is equivalent to introducing a dependence in the water discharge) (Seidl et al., 1994; Whipple and Tucker, 1999). The suitability of this model to adequately reproduce several features of bedrock channels has recently been reviewed extensively by Lague (2014). One of its main restrictions is that it does not take into account more detailed parameters such as the dynamics of the alluvial cover in the channel.

In a bedrock mountain river, various processes contribute to incision: chemical dissolution, cavitation, abrasion (or wear) by both bedload and suspended load, plucking and macroabrasion (Whipple et al., 2000; Chatanantavet and Parker, 2009). Amongst those, abrasion, plucking and macroabrasion depend directly on the amount of material that is removed from the bedrock by rolling, sliding or impacting particles transported by the flow, which itself depends mainly on the amount of en-

ergy that is transmitted to the bedrock by moving particles (Foley, 1980). This transfer of energy from the impacting particle to the bedrock has been recently directly measured experimentally by Turowski and Bloem (2015), as a function of the thickness of the sediment layer covering the bedrock. As can be expected, the fraction of the incipient kinetic energy that is effectively transmitted to the bedrock decreases when the sediment thickness increases. This confirms that, as proposed

early on by Gilbert (1877) or Shepherd (1972), the amount of sediment available in the river channel should influence the downcutting rate in two opposite ways: incision should be first enhanced by an increase in the number of impacts of abrasive tools on the bedrock ('tools effect'), but if the supply rate becomes too high, the bedrock should become partially or totally protected from these impacts ('cover effect').

The first direct measurement of the effect of sediment transport on abrasion was performed by Sklar and Dietrich (2001): by measuring the mass loss of a rock disk eroded by a bedload layer of saltating grains in a rotating flow, they confirmed that a maximum abrasion rate is observed for a critical amount of sediment above the bedrock. Following this experimental work, Sklar and Dietrich (2004) developed a mechanistic approach in order to derive the saltation-abrasion model. In this

model, the incision rate $I$ is written as the product of three terms: the volume of rock eroded by each impact, $V_i$, the number of impacts by unit time and surface, $n_i$, and the probability that a saltating grain impacts an exposed area of the bedrock, $F$:

$$I = V_i \times n_i \times F \tag{1}$$

Sklar and Dietrich (2004) derive the frequency of impacts from mechanistic description of saltation trajectories, all pebbles being assumed to have the same dynamics. They consider that $F$ is the

55 fraction of bedrock not shielded by immobile particles, and varies linearly with the amount of available sediment $Q_s$ when below the transport capacity of the stream, and vanishes when the transport





capacity $Q_t$ is reached:

$$F = \begin{cases} 1 - Q_s/Q_t & \text{for} \quad Q_s < Q_t \\ 0 & \text{otherwise} \end{cases} \tag{2}$$

A further model was developed by Turowski et al. (2007), where each saltating grain has a given probability to impact an exposed or covered region of the bedrock, which leads to an exponential

expression for $F$:

$$F = \exp\left(-\varphi\, Q_s/Q_t\right) \tag{3}$$

with $\varphi$ a constant. In both models, the bedload layer is described as made of two distinct populations: static particles that cover and protect the bedrock, and moving particles that all have the same trajectories. Therefore, this analytical approach does not take into account the fact that when the amount of sediment increases, moving pebbles will interact with each other and with static ones, which should

modify their trajectories. An impact is likely to be qualitatively different depending on whether it hits a covered or exposed region of the bedrock: a saltating particle impacting the raw bedrock can bounce and continue its saltating trajectory. On the opposite, when it impacts an area already covered with immobile pebbles, it is likely that more energy will be dissipated in the collision, and the impacting particle might not bounce back. The two populations (static and saltating particles) should

therefore be permanently interacting, with static particles being ejected by an impact and becoming mobile, while mobile particles can get trapped in asperities of the static cover (Charru et al., 2004). This implies that equation (1) is somewhat ill-defined, since the number of impacts $n_i$ should also be a function of $F$.

In this article, we propose a new numerical approach of abrasion, based on the discrete element method. This method allows us to model the individual trajectories of all particles within the bedload layer, and therefore to obtain a physically-based value of the amount of energy transmitted to the bedrock by impacts. The article in organized as follows: in section 2 we present our numerical setup and the physical laws implemented in our simulations. In section 3 we expose the numerical

results regarding the sediment transport rate, the energy delivered to the bedrock and the influence of bedrock roughness. Finally in section 4 we discuss the implications of our results on the influence of both the Shields number and the sediment supply on the incision rate. We propose a new definition for the cover function $F$, compare our results to available experimental and analytical models and estimate the long-term incision rate predicted by our simulations.





## 85  2  Description of the numerical model

### 2.1  Numerical setup

We use the Discrete Element Method to simulate the individual dynamics of pebbles entrained by a turbulent water flow over a fixed bedrock. The same method would allow to model non-spherical particles by considering composite particles made of 2 or more 'glued' spheres, but for sake of

simplicity and to limit the number of control parameters, we restrict our study to the dynamics of spherical particles.

The computational domain is a parallelepipedic box of length $L = 2\,\mathrm{m}$, width $W = 1\,\mathrm{m}$ and height $H = 2\,\mathrm{m}$ (see figure 1 and table 1 for the list of all physical parameters used in the simulation).

Periodic boundary conditions are used in both horizontal directions $x$ and $y$: any pebble coming out of the box on one side is reinjected with the same velocity on the other side. The bedrock is modelled as a horizontal surface located at $z = 0$, over which we simulate a natural roughness of the bedrock by glueing $N_b$ spheres of radius $R = 5\,\mathrm{cm}$, centered at a height $z = z_r$. These protruding spheres have the same mechanical properties as the pebbles entrained by the flow but are fixed, and

considered as part of the bedrock. In all the presented results except in section 3.4, these spheres protrude by a height $h_b = R + z_r = 4\,\mathrm{cm}$ and their surface density is given by

$$\chi = N_b \times \frac{\pi R^2}{W L} = 0.36 \tag{4}$$

Considering that we only model spherical particles, the presence of this roughness on the bedrock is necessary to allow for the existence of patches of immobile pebbles (particles roll towards asperities and can get trapped) and also to enhance vertical motion, that is, saltation of mobile pebbles

(without any roughness and within a purely horizontal flow particles tend to simply roll along the smooth surface). The aim of the simulation is to compute the amount of energy that is transmitted to the bedrock when it is hit by saltating pebbles, and to evaluate the erosion of the bedrock induced by these impacts. However, let us note that we simulate the bedload dynamics over a timescale of the order of one minute, whereas significant abrasion of the bedrock only happens over at least the du-

ration of a flood, that is, a few days, and in many cases over years. Therefore, we can admit that the bedrock (both the horizontal surface and the fixed spheres) remains unchanged and immobile within the timescale of the simulation (in particular, its altitude remains $z = 0$ throughout the whole duration of the simulation). This can be seen as an advantage compared to experimental studies such as conducted by Johnson and Whipple (2010), where incision affects preferential areas of the bedrock,

which rapidly leads to the formation of a narrow inner channel, thus making both shear stress and alluvial highly non-uniform within the whole channel.



The bedload consists of $N$ pebbles, that are modeled as spheres of radius $R = 5\,\text{cm}$ and density $\rho_s = 2{,}500\,\text{kg}\,\text{m}^{-3}$. Even if repeated impacts might lead to a slow comminution of the pebbles, their size is considered as constant over the duration of the simulation. The number of pebbles that can be disposed in a single layer over the bedrock is of the order of $WL/(\pi R^2)$. Therefore, we quantify the sediment supply by defining the dimensionless surface density $\sigma$ as the surface of bedrock covered by pebbles, divided by the total available surface:

$$\sigma = N \times \frac{\pi R^2}{W L} \tag{5}$$

A horizontal turbulent water flow in the $x$ direction puts the pebbles into motion (see section 2.3). Their trajectories are then driven by their immersed weight ($\mathbf{W}$), fluid friction (drag force $\mathbf{F}$ and torque $\mathbf{M}$) and contact forces exerted by other pebbles ($\mathbf{N}$, $\mathbf{T}$). The evolution with time of the position $\mathbf{r}$ and rotational velocity $\mathbf{\Omega}$ of a pebble is given by Newton's equations of motion:

$$\begin{cases} m\dfrac{d^2\mathbf{r}}{dt^2} &=\quad \mathbf{W} + \mathbf{F} + \mathbf{N} + \mathbf{T} \\[2mm] J\dfrac{d\mathbf{\Omega}}{dt} &=\quad \mathbf{R} \times \mathbf{T} + \mathbf{M} \end{cases} \tag{6}$$

with $m = \dfrac{4\pi}{3}\rho_s R^3$ the mass and $J = \dfrac{8\pi}{15}\rho_s R^5$ the angular momentum of a pebble. The immersed weight of a pebble is

$$\mathbf{W} = (\rho_s - \rho_w)\frac{4\pi R^3}{3}\,\mathbf{g} \tag{7}$$

with $\mathbf{g} = -g\,\overrightarrow{e_z}$ the gravitational acceleration ($g = 9.8\,\text{m}\,\text{s}^{-2}$).

## 2.2 Contacts between pebbles

When two pebbles are in contact, the exact deformation of each solid particle is not explicitly computed but spheres are instead allowed to overlap slightly (see for instance Pöschel and Schwager (2005)). We assume that two pebbles $i$ and $j$ are in collision if the distance between their centers is lower than the sum of their radii, that is, if $\delta = 2R - |\mathbf{r}_i - \mathbf{r}_j| > 0$ (see figure 2). When two pebbles make contact, they experience an inelastic rebound that can be modeled by the sum of an elastic and a viscous force (Cundall and Strack, 1979). The elastic force is linear in the overlap $\delta$. The viscous dissipation is proportional to the temporal variation of this overlap: the normal force experienced by a pebble $i$ in contact with a pebble $j$ is then

$$\mathbf{N}_{ij} = -\left(k\,\delta - \Gamma\frac{d\delta}{dt}\right)\mathbf{n}_{ij} \tag{8}$$

where $k$ is the elastic constant, $\Gamma$ is the effective viscosity and $\mathbf{n}_{ij} = \dfrac{\mathbf{r}_j - \mathbf{r}_i}{|\mathbf{r}_j - \mathbf{r}_i|}$ is the unit normal vector of the collision. The value of the elastic constant is related to the material's Young modulus and the pebble size: we adopt the value $k = 2\times10^8\,\text{N}\,\text{m}^{-1}$, which corresponds to an elastic modulus



$Y \sim k/R = 4\,\mathrm{GPa}$. This value is quite low for rocks, but increasing the elastic modulus would imply reducing too much the numerical timestep. Let us note, however, that the pebbles that we model are nevertheless very rigid: the deformation of a pebble under its own weight is only of $60\,\mathrm{nm}$. The effective viscosity is a numerical parameter that is responsible for the inelasticity of the collision but does not have a direct physical equivalent. When a pebble impacts another one in water, energy is dissipated in plastic deformations or micro-fractures within the rock (which are responsible for wear), as well as in the viscous interstitial flow. Within our model, the effective inelasticity of the collision can be quantified by the coefficient of restitution $e$, which compares the velocity of the pebble before and after a collision: $e = 1$ corresponds to an elastic collision and $e = 0$ to total dissipation. If the force is given by equation (8), $e$ is expressed as

$$e = \exp\left(-\frac{T_{coll}\Gamma}{2m}\right) \quad \text{with} \quad T_{coll} = \pi\frac{\sqrt{m/k}}{\sqrt{1 - \Gamma^2/(4mk)}} \tag{9}$$

the typical duration of a collision. We choose the value $\Gamma = 2 \times 10^4\,\mathrm{kg\,s^{-1}}$ for the effective viscosity, which leads to collisions of duration $T_{coll} = 10^{-4}\,\mathrm{s}$ and a coefficient of restitution $e = 0.3$, which means that a pebble looses $1 - e^2 = 90\%$ of its incident kinetic energy during an impact.

The tangential force $\mathbf{T}_{ij}$ generated at a contact between two pebbles is described by the regularized Coulomb's law of solid friction, as in Cundall and Strack (1979). This force opposes the tangential motion and is expressed as

$$\mathbf{T}_{ij} = -\min(G|\mathbf{v}^s|; \mu|\mathbf{N}_{ij}|)\frac{\mathbf{v}^s}{|\mathbf{v}^s|} \tag{10}$$

where $\mathbf{v}^s$ is the sliding velocity at the contact, which is a function of the two pebbles' translational and angular velocities, $\mu = 0.6$ is the local friction coefficient, $G = 5\,\mathrm{kg\,s^{-1}}$ is the slope of the regularization of the Coulomb's law. This regularization prevents the indetermination of the friction force when the two particles in contact have a zero sliding velocity.

Finally, each collision between a pebble and the horizontal surface of altitude $z = 0$ is treated as a collision with a pebble of infinite size, and same mechanical properties.

### 2.3 Turbulent water flow

A stationary turbulent flow over a rough bedrock follows the average velocity profile :

$$\mathbf{V}(z) = V_f(z)\mathbf{e}_x \quad \text{with} \quad V_f(z) = \frac{U^*}{\kappa}\ln\left(\frac{z}{z_0}\right) \tag{11}$$

where $x$ is the direction of the flow, $\kappa = 0.41$ is the von Kármán constant. $z_0$ is the bedrock roughness and depends on pebble size : a bedrock made of pebbles of radius $R$ has a roughness $z_0 = R/15$ (Nikuradse, 1933; Valance, 2005). $U^*$ is the shear velocity, whose expression is given by the relationship between turbulent shear stress and velocity gradient:

$$U^* = \sqrt{\frac{\tau}{\rho_w}} \quad \text{with} \quad \tau = \rho_w\kappa^2z^2\left(\frac{\partial V_f}{\partial z}\right)^2 \tag{12}$$



where $\rho_w = 1,000\,\mathrm{kg\,m^{-3}}$ is the density of water. The ability of the stream to put pebbles into motion is described by the Shields number. This dimensionless quantity is proportional to the ratio between the drag force on a pebble and its immersed weight:

$$\Theta = \frac{\tau \times (2R)^2}{(\rho_s - \rho_w)\,g \times (2R)^3} = \frac{\rho_w\,(U^*)^2}{2\,(\rho_s - \rho_w)\,gR} \tag{13}$$

Pebbles are put into motion by the flow if $\Theta$ exceeds a threshold value $\Theta_c$: measurements of this threshold, both in the field and in experiments, give values in the range $0.01 < \Theta_c < 0.2$ (Buffington and Montgomery, 1997; Lamb et al., 2008b). In abrasion experiments conducted in a flume setup by Attal and Lavé (2009) the maximum fluid velocity is $4\,\mathrm{m\,s^{-1}}$ for a water height $H = 60\,\mathrm{cm}$ and pebbles of size 10 to $80\,\mathrm{mm}$. As reported in data reviewed by Rickenmann and Recking (2011), flow velocity in moun-

tain streams varies typically between 0.3 and $4\,\mathrm{m.s^{-1}}$, for a water height between 0.1 and $3\,\mathrm{m}$. In order to be consistent with these observations, in our simulations we adopted mean water velocities up to $5.0\,\mathrm{m\,s^{-1}}$, which corresponds to $\Theta$ varying from 0 to 0.11.

If the bedrock is covered with a layer of mobile pebbles, as in our simulations, the turbulent ve-

locity profile is modified. Recently, Duran et al. (2012) developed a quasi-2D mechanistic approach that takes into account the retroaction of the pebbles on the water flow, by assuming the conservation of total horizontal momentum in horizontal slices. They showed that the fluid velocity vanishes where the local solid fraction is high enough (that is, at a depth of one or two grain diameters within the bedload layer), and tends to a logarithmic profile in the 'clear water' region. The 'intermediate'

region where the velocity goes from zero to the logarithmic profile is very thin (usually of the order of one grain diameter). Therefore, we simplified the treatment of this retroaction by modeling only two different regions in the flow: at each timestep, we compute the average solid fraction $\phi$ in horizontal slices. If $\phi < \phi_b = 0.5$, the velocity profile is logarithmic and not affected by the presence of pebbles. If $\phi \geqslant \phi_b$, the velocity of water vanishes : $V_f = 0$ (see the resulting velocity profile in

figure 5). This approximation would be too simplistic if we were to study the exact flux of grains at the surface of a thick layer, but remains relevant enough in our simulations where in most cases the bedload layer remains relatively thin. A more detailed description in our geometry would require not only to take into account the average retroaction of grains on the fluid flow, but also the horizontal variations in pebble density.

## 2.4 Interactions between pebbles and the flow

The turbulent flow exerts on each mobile pebble a drag force given by

$$\mathbf{F} = \rho_w \frac{\pi R^2}{2} C_D\,|\mathbf{U}|\,\mathbf{U} \tag{14}$$

where $\mathbf{U} = \mathbf{V}(z) - \dfrac{d\mathbf{r}}{dt}$ is the relative velocity between the local flow and the pebble. The drag coefficient $C_D$ of a sphere can be expressed semi-empirically as a function of the particle Reynolds



number (Clift et al., 1978). In the present study, we use the following approximation:

$$C_D = \frac{24}{\text{Re}_p} + 0.4 \tag{15}$$

where $\text{Re}_p = \dfrac{2\rho_w U R}{\eta_w}$, with $\eta_w = 10^{-3}\,\text{Pa s}$ the dynamic viscosity of water, is the particular Reynolds number. This approximation is equivalent to the Stokes formula for the drag force at low $\text{Re}_p$. When a sphere is rotating in a viscous fluid such as water, its angular velocity induces a diffusion of momentum in a boundary layer. This results in a viscous torque applied to the pebble (Liu and Prosperetti, 2010), which opposes its rotation:

$$\mathbf{M} = -8\pi\eta_w R^3 \mathbf{\Omega} \tag{16}$$

**2.5   Computational methods**

Pebbles are initially disposed on a regular lattice at a height $z = 8\,\text{cm}$ and released with no initial velocity at $t = 0$. At the same time, the fluid is set into motion and pebbles start to move, driven by both gravity and the drag force. We use the classical numerical methods of Molecular Dynamics to compute the positions ($\mathbf{r}$) and rotational velocities ($\mathbf{\Omega}$) of the pebbles as a function of time: at each

timestep, all forces acting on each pebble are computed, and Newton's equations of motion (both translational and rotational) are integrated simultaneously for all pebbles by the Verlet method, of fourth order (Cundall and Strack, 1979; Pöschel and Schwager, 2005).

The timestep used in the simulation is $\Delta t = 10^{-6}\,\text{s} = T_{coll}/100$, which ensures that the trajectories during a collision are computed with sufficient accuracy. The 'instantaneous' sediment flux

$q_s(t)$ is computed over temporal windows of duration $\delta t = 100\,\text{ms}$:

$$q_s(t) = \frac{1}{\delta t} \times \frac{1}{WL} \times \int\limits_{t}^{t+\delta t} \left( \sum_{i=1}^{N} m\, v_x{}^i(t') \right) dt' \tag{17}$$

Within the alluvial cover, some pebbles are almost immobile, either because they got trapped by the bedrock roughness, or because they belong to bottom layers of the cover and are therefore not entrained by the water flow. These pebbles constitute a static cover, that contributes to protect the bedrock from rapid impacts by saltating pebbles. We quantify the cover fraction in the following

way: the bedrock surface is divided into square cells of side $2R$. At each timestep, we compute the velocity distribution of pebbles. If a pebble centered in a given cell has a velocity lower than 1/10th of the maximum velocity, this cell is considered as 'covered' by an immobile pebble. If $n$ cells are covered at a time $t$, the time-averaged static cover fraction is then defined as

$$C = \langle n(t) \rangle \times \frac{4R^2}{WL} \tag{18}$$

With this definition, $C = 1$ when one layer of immobile particles is completely shielding the bedrock.

As are collisions between mobile particles, each impact of a pebble on the bedrock is inelastic: the impacting pebble looses a fraction of its incipient kinetic energy during the collision (due to the



dissipative term in equation (8)). This energy loss can result in the erosion of a small volume of bedrock (see section 4.4). We can evaluate the energy lost during an impact by computing the work of the repulsive force during the collision, that is

$$\Delta E = \int\limits_{\text{collision}} \mathbf{N}(t') \cdot \delta(t') \, \mathbf{n} \, dt' \tag{19}$$

If we consider all impacts on bedrock occurring over a duration $T$, the total energy delivered to the bedrock by unit time and surface can then be expressed as

$$\Phi_E = \frac{1}{WL} \times \frac{1}{T} \times \sum_{\text{impacts}} \Delta E \tag{20}$$

## 3  Results

### 3.1  Sediment transport

Let us first investigate the structure and dynamics of the bedload layer. Figure 3 shows the evolution of the flux of sediment $q_s$ with time when the Shields number is beyond the threshold of motion. After the pebbles are released in the flow, the bedload flux increases regularly during a transient phase. The duration of this transient phase depends on the Shields number but is always of the order of a few seconds. The bedload flux then reaches a steady value: from this moment we consider that the system is in the permanent regime. In the rest of the discussion all results are computed in this permanent regime only.

Transport of sediment only occurs if the fluid drag force on a pebble is large enough to overcome solid friction, that is, if the Shields number exceeds a critical value $\Theta_c$. In figure 4 we plot the flux of sediment, averaged over time (in the permanent regime) $Q_s = \langle q_s(t) \rangle$ as a function of $\Theta$. We observe that the threshold of motion corresponds to a critical value of the Shields number $\Theta_c = 0.012$. Below this threshold, the average sediment flux vanishes after a short transient. The exact value of the threshold is somewhat difficult to assess, since some transport can occur with intermittency even below $\Theta_c$. Within the margin of error of this definition, the threshold $\Theta_c$ depends only very slightly on the sediment supply $\sigma$ and we shall assume in the following that $\Theta_c$ is a constant. The value found in our simulations is relatively low compared to many experimental observations. However, let us note that our particles are perfectly spherical and therefore easy to put into motion, plus most observations of the transport threshold concern the incipient motion of particles over a thick sediment layer, where isolated moving particles are more likely to get trapped and stop their motion at low Shields number.

In figure 5, we plot the flow velocity, the mean velocity of pebbles and the solid volume fraction as a function of height and for a relatively large sediment supply ($\sigma = 1.6$). The local volume fraction is computed in horizontal slices of height $R/3.5$. Its profile presents two local maxima and vanishes for $z \sim 2R$, which shows that the bedload is structured (in this example) into two rather compact



layers. As evidenced by the water velocity profile, the flow only penetrates the upper layer of pebbles
(which in this case is incomplete). Most of the pebbles lying in the bottom layer are therefore totally
immobile or only slightly entrained by the upper mobile particles. The average velocity of pebbles
exposed to the flow increases with their vertical position in the bedload, but remains lower than the
velocity of water.

Let us now investigate the transport law by varying the Shields number $\Theta$. In figure 6, we plot the
270 variation of the average flux of sediment $Q_s$ with the reduced Shields number $(\Theta - \Theta_c)/\Theta_c$, and for
different values of the sediment supply. The classically used Meyer-Peter/Müller law (Meyer-Peter and Müller,
1948)

$$Q_{sat} = 8\rho \sqrt{\frac{\rho - \rho_w}{\rho_w} g (2R)^3} \times (\Theta - 0.047)^{3/2} \qquad (21)$$

is plotted for comparison. By analogy with most sediment transport laws, which give the flux of
sediment at saturation, we fit the evolution of $Q_s$ with $(\Theta - \Theta_c)$ by a power law, whose prefactor
depends on the amount of sediment available, that is, on the sediment density $\sigma$:

$$Q_s(\sigma, \Theta) = f(\sigma) (\Theta - \Theta_c)^{n(\sigma)} \qquad (22)$$

It has to be noted, though, that we only explore a limited range of Shields number, considering that
the sediment supply $\sigma$ is the main control parameter in this study, which does not garantee a high
precision in the determination of the index $n$. The index of the best fit roughly increases with the
sediment supply, and we have $n(\sigma) < 1$ for $\sigma < 1$ and $n(\sigma) > 1$ otherwise. If the sediment supply is
280 low, the sediment flux increases slowly with $\Theta$: pebbles can be transported at a higher speed when
the flow accelerates, but the amount of available pebbles remains below the transport capacity. If the
sediment supply is high enough, a rapid flow is able to put more pebbles into motion, which leads to
a rapid variation of $Q_s$ with $\Theta$.

Let us now focus on the effect of the sediment supply on the bedload flux. In figure 7a), we plot
the flux of sediment as a function of the sediment supply for a few values of the Shields number.
For a given stream velocity, the amount of available pebbles and therefore the sediment flux both
increase with $\sigma$. However, if the total number of pebbles is too high, the transport capacity of the
flow is reached and the flux of sediment saturates. Let us remark that in most cases a local minimum
in the bedload flux is reached around $\sigma = 1$, which can be seen both as a geometrical effect (mobile
pebbles can then form a compact layer, consolidated by the bedrock roughness, and become hard to
dislodge) and as an artificial effect of our fluid model: if the volume fraction in the only bottom layer
is low, the fluid velocity is larger than zero; it vanishes as soon as the first layer is dense enough. If
the sediment supply slightly increases, some pebbles will pop up above the first layer and be easily
entrained by the flow. Similar but less pronounced local minima can also be observed around $\sigma = 2$
and $\sigma = 3$.

Figure 7b) confirms that the function $f(\sigma) = \dfrac{Q_s}{(\Theta - \Theta_c)^{n(\sigma)}}$, though not monotoneous, is indeed
independent of $\Theta$. The variation of $Q_s$ with $\sigma$ shows that the sediment flux first increases linearly



before saturating beyond a critical sediment supply $\sigma_0$. For sake of simplicity and in order to evaluate this critical value, we dismiss the local minima reached at $\sigma = 1, 2$ and $3$ and fit the curve $Q_s(\sigma)$ by a simple exponential function:

$$Q_s(\Theta, \sigma) = Q_t(\Theta) \left(1 - e^{-\sigma/\sigma_0}\right) \tag{23}$$

Given this model, the surface density $3\sigma_0$ corresponds to the maximum quantity of sediment that can be transported by a flow of given Shields number. The variation of $\sigma_0$ with $\Theta$ is plotted in figure 7c): it can be well fitted by the affine function

$$\sigma_0 = 20(\Theta - \Theta_c) \tag{24}$$

This finding is once again consistent with most transport models in the limit of high sediment supply: if the amount of sediment is large enough, the number of particles put into motion increases linearly with $\Theta$, whereas their velocity is proportional to the shear velocity, that is, to $\Theta^{0.5}$. Finally in figure 8 we plot the maximum sediment transport rate $Q_t(\Theta)$, reached for large values of the sediment supply $\sigma \gg \sigma_0$. The best fit by a power law suggests $Q_t \sim (\Theta - \Theta_c)^{1.2}$. However, let us note that we do not have many data points, and that modeling spherical particles is likely to enhance transport close to the threshold. As shown in the same figure, our data could also be consistent with the power law $Q_t \sim (\Theta - \Theta_c)^{1.5}$.

### 3.2 Static cover

In figure 9, we plot the evolution of the static cover fraction $C$ as a function of $\sigma$ for different values of the Shields number. As expected, below the threshold of motion, the static cover fraction first increases linearly with the sediment supply. Because of the roughness of the bedrock, it departs from the function $C = \sigma$ beyond $\sigma \simeq 0.5$: this is due to the fact that the distribution of pebbles on the surface is not strictly homogeneous. A fraction of the bedrock is then covered by two layers of immobile pebbles while other areas are bare. If $\Theta$ exceeds the threshold of motion, the immobile cover fraction is very low for $\sigma < 1$ (and strictly zero when the Shields number is high enough, all particles being entrained by the flow), where mobile pebbles have the freedom to roll along the bedrock. If the sediment supply increases, an incomplete static layer develops over the bedrock. In all cases the bedrock becomes entirely covered by a static layer if the sediment supply exceeds $\sigma \simeq 3$. Local maxima in the static cover fraction are, once again, due to the intrinsic discontinuity of our fluid model: when the bottom volume fraction exceeds $\phi_b = 0.5$, the fluid velocity suddenly decreases within the bottom layer.

### 3.3 Incision process

The flux of energy that is delivered to the bedrock by the impacts is given by the work of the dissipative normal force during each collision between a mobile pebble and the bedrock (whether it is the





flat surface or one of the glued spheres). In figure 10, we plot the variation of this flux of energy $\Phi_E$
with $\Theta$ for different values of the sediment supply. As can be expected, $\Phi_E$ becomes larger when
the velocity of the stream increases: if pebbles move at a higher speed in the bedload layer, their
incoming velocity at the impact on the bedrock increases, as does the energy dissipated during the
impact. As illustrated in figure 10a, the variation of $\Phi_E$ with $\Theta$ can be fitted by a power law whose

index varies with the sediment supply:

$$\Phi_E(\Theta, \sigma) = \Phi_1(\sigma)(\Theta - \Theta_c)^{m(\sigma)}. \tag{25}$$

As shown in figure 10b, the exponent $m(\sigma)$ increases roughly linearly with $\sigma$, with $m(0) \simeq 1$ and
more generally $1 \leq m(\sigma) \leq 4$.

   Let us now investigate more precisely the effect of the sediment supply on the energy transfer,

which is plotted in figure 11 for a few values of the Shields number. The shape of the curve is similar
in all cases: $\Phi_E$ first increases with $\sigma$ until it reaches a maximum value, and then decays to zero for
large sediment supplies. The estimate of the incision rate corresponding to a given flux of energy
delivered to the bedrock is obtained though the procedure described in section 4.4.

   The insert in figure 11 shows the variation of the maximum flux of energy with the Shields number.

This variation can be well fitted by an affine relationship. This demonstrates that the process of inci-
sion only happens if the Shields number exceeds an incision threshold $\Theta_i \simeq 0.025 > \Theta_c$. Therefore,
sediment transport can occur on a bedrock while not contributing to river incision if $\Theta_c < \Theta < \Theta_i$: in
this regime, pebbles are rolling along the bedrock without impacting, and therefore do not contribute
significantly to its erosion.

### 350    3.4   Influence of bedrock roughness

   The roughness of the bedrock can be modified by varying two parameters: the surface density $\chi$ of
fixed spheres on the bedrock and the protruding height of those pebbles over the horizontal bedrock
(roughness height $h_b = R - z_r$). Figure 12 illustrates the aspect of the bedrock, viewed from above,
for two different values of $\chi$.

In order to study the variation of energy transfer with the bedrock roughness, we plot the flux
of energy delivered to the bedrock with respect to the sediment supply ($\sigma$) for a single value of
$\Theta - \Theta_c = 0.071$, and for different roughness configurations. Figure 13a) shows the same plot for
increasing values of $\chi$, and a roughness height $h_b = 4\,\mathrm{cm}$. The aspect of the erosion curve is the
same whatever the roughness density. However, a higher density of protruding spheres systematically

leads to a decay of the energy received by the bedrock at high sediment supplies: the cover-effect is
more efficient if the bed possesses a dense roughness. In figure 13b), we plot the same evolution for
a given roughness density $\chi = 0.36$ but for spheres protruding more or less within the flow. At high
sediment supply, the energy received by the bedrock appears to be higher if the bedrock is smooth.





## 4 Discussion

### 4.1 Sediment transport rate

Although the model that we adopt for the interaction between the water flow and the pebbles is rather simple, the dynamics of the bedload layer appears to be consistent with experimental observations. As shown by figure 6, the order of magnitude of the sediment transport rate is comparable to the one predicted by the Meyer-Peter/Müller law. When the transport rate is fitted by a power law, the exponent that we obtain depends on the dimensionless sediment supply $\sigma$ and is in general smaller than $1.5$. At low Shields number, the sediment flux is higher for low supplies: indeed, it is easier for pebbles to roll along a flat bedrock than on the rough surface of a sediment layer. When $\Theta$ is increased, the bedload flux increases faster for large sediment supply. For low supplies, once all pebbles have been put into motion, only their velocity can increase with $\Theta$. If more pebbles are available, an increase in $\Theta$ leads to both more pebbles moving, and an increase of their velocity, which implies $n(\sigma) > 1$. The main discrepancy between our results and experimental observations is the value of the motion threshold, which is much lower ($\Theta_c \simeq 0.012$) in our simulations than in most reported measurements (Lamb et al., 2008a). This is likely due to the fact that we model pebbles as spheres, which can easily roll along the bedrock. Let us note that with only spheres as well but modeling more sophisticated fluid-grain interactions, Duran et al. (2012) obtain a critical Shields number of $0.12$. However, like in most experimental studies they are interested in the threshold of motion for grains rolling along the surface of a thick sediment layer, rather than on a relatively smooth surface, as is the case in our simulations. It would be necessary to consider more irregular pebbles (for instance by considering cohesive clusters of several spherical particles) in order to perform more realistic simulations. Once rescaled by the value of the threshold, the sediment flux that we obtain is, though, quantitatively consistent with common measurements.

### 4.2 The role of sediment supply, Shields number and bed roughness on incision

As shown in figure 11, the results of our simulations are qualitatively consistent with experimental observations by Sklar and Dietrich (2001): for a given Shields number $\Theta$, energy dissipated in the bedrock first increases with the sediment supply and reaches a maximum for $\sigma_m \simeq 0.5$ before decaying, and vanishing at high sediment supply. This can be understood as the result of a competition between a 'tool-effect' and a 'cover-effect': as long as the bedrock is still exposed to impacts, the more pebbles are put into motion, the more energy they provide to the bedrock. When the sediment supply increases, immobile (or slowly rolling) pebbles start to accumulate on the bedrock, thus partially protecting it from direct impacts by saltating pebbles. The total energy transferred to the bedrock vanishes totally beyond $\sigma \simeq 3$: at this point the bedrock is totally protected by the bedload layer. In figure 14, we compare more quantitatively our numerical results to the experiments by Sklar and Dietrich (2001) and the predictions of the linear (Sklar and Dietrich, 2004) and expo-





nential (Turowski et al., 2007) cover models. Since these results were originally expressed in terms
of the mass $m_s$ of gravel of diameter $d$, within a cylindrical container of diameter D, we derive the
corresponding dimensionless sediment supply through the expression

$$\sigma = N \times \frac{\pi d^2/4}{\pi D^2/4} = \frac{6\,m_s}{\rho_s\,\pi\,d\,D^2} \tag{26}$$

The comparison shows that our simulations are able to predict the right tendency for the flux of
energy delivered to the bedrock, that is, for the incision rate. However, let us note that because of
the cylindrical geometry of the experiment by Sklar and Dietrich (2001), the fluid shear stress and
the cover fraction are not uniform in their setup: immobile particles tend to accumulate in the cen-
ter of the disk, while saltating grains can still impact the bedrock around this protected area. This
discrepancy might explain why the value of $\sigma$ corresponding to maximum incision differs slightly
between our simulations and the experiments. The quantitative description of the cover effect will
be discussed further in section 4.3.

We also quantified the influence of the Shields number on the abrasion process and showed a
power-law dependency of $\Phi_E$ on the excess shear stress $\Theta - \Theta_c$. In particular, our fits (see fig-
ure 10b) show that the exponent $m$ is always greater than the index $n$ of the transport law. This
implies that if $Q_s$ and $\sigma$ are kept constant, the incision rate increases with the Shields number.
This result contradicts the prediction of the saltation-abrasion model where the incision rate scales
like $(\Theta - \Theta_c)^{-0.5}$ (Sklar and Dietrich, 2004), and findings by Chatanantavet and Parker (2009) and
Johnson and Whipple (2010) where there is no explicit dependency of $\Phi_E$ in $\Theta$ (for given values of
$Q_s$ and $\sigma$). However, it is consistent with the prediction of the shear stress and stream power incision
models (Whipple and Tucker, 1999).

Our results also show that abrasion only occurs beyond a given threshold which is higher than the
threshold of motion of pebbles, which can be explained by the fact that rolling or sliding pebbles do
not contribute significantly to the erosion of bedrock. This is inconsistent, however, with observa-
tions by Sklar and Dietrich (2001) who report observing abrasion as soon as the flow is able to put
sediment into motion. Let us note, however, that we vary the flow velocity for a given pebble size
whereas Sklar and Dietrich (2001) vary the sediment size for a given velocity. The fact that there is
only a small difference between the two values $\Theta_c$ and $\Theta_i$, could explain that the discrepancy was
not observed experimentally. If the existence of an incision threshold $\Theta_i > \Theta_c$ was confirmed, it
would mean that this value should be taken into account in models of river incision instead of the
critical Shields number $\Theta_c$. This is also consistent with the possible existence of an energy thresh-
old necessary to effectively erode a small volume of material at impact, as observed by Bitter (1963).



Our simulations show that the roughness of the bedrock does not affect the general evolution of
the energy delivered to the bed with respect to the cover fraction. However, incision appears to be
enhanced if the surface density of asperities is low and if they are not too high. These two effects
can be related to the geometrical explanation of the cover effect: if the roughness is denser or higher,
mobile pebbles are more likely to get trapped and immobilized along the bedrock, therefore pro-
tecting it from further impacts by rapid pebbles. This enhanced cover effect will disappear it the
roughness density $\chi$ is too large: indeed, if the bedrock was entirely covered with glued spheres,
it would become equivalent to a smooth bedrock. Let us note that recently Huda and Small (2014)
modified the saltation-abrasion model in order to take into account bedrock roughness, and found the
opposite result: the incision rate is considerably increased (by more than one order of magnitude) by
the presence of long-scale bed topography. However, the roughness that we implement in our model
does not modify the local slope, and has a lengthscale comparable to the pebble size.

Finally, let us remark that the influence of the coefficient of restitution on the results of our simu-
lations should be of importance and will be the object of further investigation. Increasing the coeffi-
cient of restitution would certainly facilitate the saltating motion of pebbles, whereas they only roll
along the bedrock at low $e$. Increasing $e$ could therefore decrease the incision threshold, by narrow-
ing the rolling/sliding regime. Besides, a high coefficient of restitution means that a lower fraction
of the kinetic energy of the projectile is delivered to the bedrock. However, this implies that the im-
pacter rebounces with a higher kinetic energy, and is then more likely to impact again the bedrock at
high speed. It is therefore not trivial to assess in which way the total energy delivered to the bedrock
(that is, the number of impacts multiplied by the energy given at each impact) will evolve with the
value of $e$.

### 4.3 Cover effect

By analogy with both the linear (Sklar and Dietrich, 2004) and the exponential (Turowski et al.,
2007) cover models, we first isolate in $\Phi_E$ the influence of the sediment supply, and fit the flux of
energy by the empirical function

$$\Phi_E(\Theta, \sigma) = \Psi(\Theta)\sigma^p e^{-\sigma/\gamma}. \tag{27}$$

The values for $p$ and $\gamma$, corresponding to the best fits plotted in figure 11, are reported in table 2.
The parameter $\gamma$ appears to be roughly independent of the Shields number, and we find consistently
$p > 2$, whereas $p = 1$ in the exponential cover model. The fact that $p$ depends on $\Theta$ underlines that
we cannot express the incision rate as a simple product of a function of $\Theta$ and a function of $\sigma$.
Besides, in our simulations, forcing a fit with $p = 1$ always leads to underestimate the maximum
incision rate and overestimate it at high sediment supply. This is likely due to the fact that, in the
saltation-abrasion model, the number of impacts is proportional to the sediment supply, whereas our



simulations show that the sediment flux does not vary linearly in $\sigma$. Therefore, the incision rate increases faster than linearly at low sediment supply.

Following the approach for incision rate by Sklar and Dietrich (2004) and Turowski et al. (2007) (see equation (1)) and for the impulse rate by Turowski and Rickenmann (2009), let us express the flux of energy delivered to the bedrock, defined by equation (20), as the product of the energy provided by each impact ($E_i$), the number of impacts per unit time and surface ($n_i$) and a cover function

($F$):

$$\Phi_E = E_i \times n_i \times F \tag{28}$$

In the saltation-abrasion model, $n_i$ is proportional to the sediment supply while the cover function $F$ is interpreted as the probability that an impact hits the bedrock. In the original model by Sklar and Dietrich (2004), $F$ is simply the fraction of exposed bedrock and decays linearly with the relative sediment flux $Q_s/Q_t$. In the stochastic model of Turowski et al. (2007), $F$ is expressed as

an exponential function of $Q_s/Q_t$, leading to a prediction closer to experimental observations (see figure 14). As noted in the introduction, this equation implies that the number of impacts and the cover function $F$ are independent parameters. However, if we take into account the fact that mobile pebbles are not isolated from one another, nor from the static cover, they should be interdependent. Let us now reformulate equation (28) in order to take into account only the Shields number and

the sediment flux, which are easier to compute, or to measure experimentally, than the frequency of impacts and their energy. Indeed, if a saltating particle hits a mobile particle shielding the bedrock, a small fraction of its incipient energy could still be transmitted to the bedrock (Turowski and Bloem, 2015), though this event would not be counted as an eroding impact with equation (1). From the simulations we can then compute a cover function $F(\sigma, \Theta)$ that does not require a geometrical or

stochastical description of the alluvial cover of the bedrock. The energy of an impact $E_i$ is expected to scale like the typical kinetic energy of moving pebbles, which is itself proportional to $U^{*2}$. Using equation (13), and assuming that the impact energy vanishes for $\Theta < \Theta_c$, we can therefore write, dimensionally,

$$E_i \sim (\rho_s R^3) \times U^{*2} \sim (\rho_s R^3) \times g\,R(\Theta - \Theta_c) \tag{29}$$

As proposed by Foley (1980), the number of impacts per unit time and surface is expected to be

proportional to the sediment transport rate $Q_s$, which can be written dimensionally as

$$n_i \sim \frac{Q_s}{\rho_s R^4} \tag{30}$$

Therefore, the flux of energy should read:

$$\Phi_E = K \times g \times Q_s(\sigma, \Theta) \times (\Theta - \Theta_c) \times F(\sigma, \Theta) \tag{31}$$





with $K$ a dimensionless constant, which depends neither on $\Theta$ nor on $\sigma$. This allows us to redefine the cover function as

$$F(\sigma, \Theta) = \frac{\Phi_E}{g\, Q_s\, (\Theta - \Theta_c)} \qquad (32)$$

In figure 15 we plot $F$ in a log-linear scale. Let us first observe that $K$ is indeed independent on $\Theta$, since all curves converge to $F = 1$ for $\sigma = 0$, which implies that $K = 1$ and validates our dimensional analysis. On one hand, as emphasized in the insert of figure 15, the cover function $F$ decays slower than exponentially, and rather linearly, for $\sigma \lesssim 0.75$. On the other hand, $F$ can be well fitted by an exponential decay for $\sigma \gtrsim 1$. This behavior is similar to the case that Hodge and Hoey (2012) refer to as 'sigmoidal' in their cellular automaton model (though as a function of $Q_s/Q_t$ and not $\sigma$, see their figure 7, and although they only consider static cover), and to some observations by Chatanantavet and Parker (2008), where the cover function appears to decay below 1 only for $Q_s/Q_t \gtrsim 0.25 - 0.75$ (see their figure 13). Our result therefore confirms that the 'exponential cover model' overestimates the cover effect at low sediment supplies but fits correctly at high enough sediment supplies. In the exponential regime, following the stochastic approach by Turowski et al. (2007), we can fit the cover function by

$$F = A \exp(-\varphi\, \frac{\sigma}{3\sigma_0}) \qquad (33)$$

with $3\sigma_0$ an estimate of the normalized sediment mass transport capacity (see figure 11) and $\varphi$ the 'cover factor' of the probabilistic approach (Turowski et al., 2007). Our results show systematically $\varphi > 1$ (see table 3), which implies that it is more probable for a pebble to impact on an uncovered zone of the bedrock than a covered one. This is consistent with observations by Chatanantavet and Parker (2008) (in flume experiments) and Turowski and Rickenmann (2009) (in the field). In figure 15, we also plot the function $1 - C(\sigma)$, where $C$ is the static cover fraction computed in section 3.2. If the evolution of both functions with $\sigma$ is similar, the cover function $F$ is systematically smaller than $1 - C$: this implies that the cover effect is not only due to immobile pebbles but also to mobile (rolling or saltating) pebbles. The latter can either directly shield the underlying bedrock (which was referred to as 'dynamic cover effect' by Turowski et al. (2007)) or hit other saltating pebbles and slow them down.

### 4.4 Estimation of the incision rate

We can estimate the rate of incision induced by the impacts on the bedrock, based on the flux of energy delivered. Following Engle (1978) and Sklar and Dietrich (2004), we express the incision rate as

$$I = \frac{\Phi_E}{\epsilon_v} \quad \text{with} \quad \epsilon_v = k_v \frac{\sigma_T^2}{2Y} \qquad (34)$$

$\epsilon_v$ being the energy required to incise a unit volume of rock. The value that is used in Sklar and Dietrich (2004) is derived from the mechanical properties of rocks: $\sigma_T = 7 \times 10^6$ Pa is the rock tensile





strength, $Y = 5.0 \times 10^{10}$ Pa is the rock elastic modulus and $k_v = 10^6$ is a dimensionless rock re-
sistance parameter. The incision rate obtained is plotted as a function of the dimensionless sed-
iment supply and for different values of the Shields number in figure 11. For instance, the inci-
sion rate corresponding to the value $\Phi_E = 40\,\mathrm{W.m}^{-2}$ (obtained at $\Theta - \Theta_c = 0.049$ and $\sigma = 0.6$) is
$I = 2.6\,\mathrm{m\,yr}^{-1} = 7\,\mathrm{mm.day}^{-1}$. Let us note that this value corresponds to an instantaneous incision
rate and not to the average incision rate over a year: it may be reached for a high water discharge and
for a particular value of the sediment supply. In a river, these conditions may be verified only during
a few days per year, while the instantaneous incision rate would be very low the rest of the time,
when the discharge is small, and the sediment supply is either very low or very high. The order of
magnitude of the incision rate that we predict is comparable to values measured in rapidly eroding
rivers in Taïwan during storm events (Hartshorn et al., 2002). As we have predicted the value of the
instantaneous incision rate for a wide range of both Shields number and sediment supply, it would
be possible to compute the long-term average incision rate for a given stream, provided that the
probability distribution functions of both water discharge and sediment supply are known.

## 5    Conclusions

We have presented the results of a new model for incision of a river bedrock based on the direct
simulation of physically based trajectories of pebbles in a stream. In this model we solved the equa-
tions of motion for a large number of pebbles entrained by a turbulent water flow, with a simplified
retroaction of the presence of the pebbles on the flow. This allowed us to compute explicitly the
trajectories of pebbles transported by the flow, and therefore to quantify the energy dissipated during
collisions between the bedload and the bedrock, which is directly responsible for the incision of
the bedrock. We found that the sediment transport rate can be fitted by a power law of the Shields
number, similarly to most classical transport laws at saturation. However, we also evidenced the
influence of the sediment supply: the exponent of the transport law increases with the quantity of
available pebbles. For a given Shields number, we showed that the bedload flux increases with the
sediment supply until it reaches its saturated value. This allowed us to compute the sediment mass
that the flow is able to transport. However, extracting a unique general expression for the flux of
sediment as a function of both the Shields number and the sediment supply remains not trivial.

The amount of energy that impacts of saltating pebbles deliver to the bedrock can be directly
computed from the simulation data. This flux of energy, which is expected to be proportional to the
incision rate, shows the same qualitative variations with sediment supply as observed in experiments
by Sklar and Dietrich (2001): it first increases with the amount of sediment available (as the number
of impacts increases) before decaying when there is too much sediment and the bedrock becomes
shielded. We also showed that the energy delivered to the bedrock increases as a power law of the



Shields number, and is zero below a given incision threshold, higher than the motion threshold,
which was not observed in experiments. Finally, by extracting a cover function from our data, we
showed that the classical linear and exponential models for the cover effect lead to underestimate
the incision rate, respectively at high and low sediment supplies. The shape of our cover function
resembles experimental observations by Chatanantavet and Parker (2008) and some numerical re-
sults by Hodge and Hoey (2012). If defined as in equation (33), the cover function appears instead
to decay linearly at low sediment supply and exponentially at high sediment supply. This underlines
the fact that the amount of sediment available contributes not only to shield the bedrock but also to
change the dynamics of saltating particles. Finally, we evaluated the rate of incision predicted by
our simulations as a function of the hydraulic conditions (the Shields number) and the amount of
sediment available (dimensionless sediment supply). Its order of magnitude appears to be consistent
with long-term observations made in mountain streams.

Though our results are qualitatively consistent with experimental observations and another type
of models, the quantitative aspect is probably affected by our numerical method: indeed the fact that
we model the flow by a horizontally-averaged and purely horizontal velocity profile is likely, on one
hand, to have a (negative) impact on the possibility for pebbles to gather into immobile patches, and
therefore on the efficiency of the cover effect. On the other hand, taking into account turbulent ve-
locity fluctuations, and in particular local bursts of vertical velocity, could enhance saltating motion.
A better explicit model of the dynamics of the alluvial cover would therefore require to account for a
spatially non-uniform velocity field, and ideally the exact velocity field around each mobile particle,
which would be much more time-consuming numerically. Such stochastic effects are probably better
accounted for in models such as the cellular automaton by Hodge and Hoey (2012). Finally, let us
note that in the prediction of the long-term evolution of a river bed (see for example (Lague, 2010)),
incision of the bedrock is not the only relevant parameter. Our numerical approach would also be rel-
evant for the study of the incision of lateral walls (if we add sidewalls to the computational domain)
and the comminution of mobile particles (since our simulations also give us access to the energy lost
by mobiles pebbles, not only in their impacts with the bedrock but also in all their contacts with one
another).

*Acknowledgements.* Data supporting the results of this article can be obtained by contacting Vincent Langlois
(vincent.langlois@univ-lyon1.fr).
The authors would like to acknowledge the insightful input of four reviewers, amongst whom P. Chatanan-
tavet, on an earlier version of this article.




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



| | |
|---|---|
| $L$ | length of the box [m] |
| $W$ | width of the box [m] |
| $H$ | height of the box [m] |
| $N$ | number of mobile pebbles |
| $R$ | radius of pebbles [m] |
| $\rho_s$ | pebble density [kg m$^{-3}$] |
| $g$ | gravitational acceleration [m s$^{-2}$] |
| $k$ | elastic constant of collisions [kg s$^{-2}$] |
| $\Gamma$ | effective viscosity of collisions [Pa s] |
| $e$ | coefficient of restitution of collisions |
| $\delta$ | overlap between two pebbles [m] |
| $\mu$ | local friction coefficient |
| $\sigma$ | normalized sediment supply |
| $\sigma_0$ | movable sediment supply at a given Shields number |
| $\sigma_m$ | sediment supply for maximal incision rate |
| $\phi$ | solid volume fraction |
| $\tau$ | shear stress [Pa] |
| $U^*$ | shear velocity [m s$^{-1}$] |
| $\kappa$ | von Kármán constant |
| $\rho_w$ | density of water [kg m$^{-3}$] |
| $\eta_w$ | dynamic viscosity of water [Pa s] |
| $\Theta$ | Shields number |
| $\Theta_c$ | critical Shields number for transport of sediment |
| $\Theta' = \Theta - \Theta_c$ | dimensionless excess shear stress |
| $\Theta_i$ | critical Shields number for incision of the bedrock |
| $N_b$ | number of spheres glued on the bedrock |
| $h_b$ | height of the bedrock roughness [m] |
| $\chi$ | surface density of bedrock roughness |
| $T_{coll}$ | typical duration of a contact between 2 pebbles [s] |
| $\Delta t$ | timestep in the simulations [s] |
| $q(t)$ | instantaneous sediment flux [kg m$^{-1}$ s$^{-1}$] |
| $Q_s$ | average sediment flux [kg m$^{-1}$ s$^{-1}$] |
| $\Phi_E$ | flux of energy [W m$^{-2}$] |
| $I$ | incision rate [m s$^{-1}$] |
| $k_v$ | dimensionless rock resistance coefficient |
| $Y$ | Young's modulus of rock [Pa] |
| $\epsilon_v$ | energy required to erode a unit volume of rock [J m$^{-3}$] |
| $\sigma_T$ | tensile strength of rock [Pa] |

**Table 1.** List of the physical parameters used in the model.




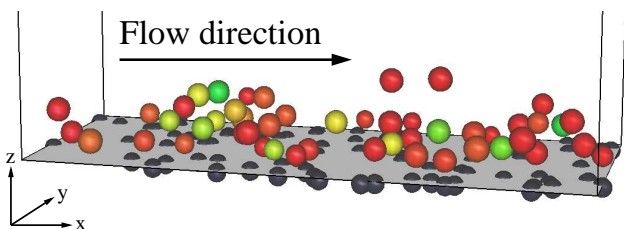

**Figure 1.** Snapshot of the numerical simulation. The color scale codes the horizontal velocity of each pebble. The grey plane and (immobile) grey spheres constitute the rough bedrock.

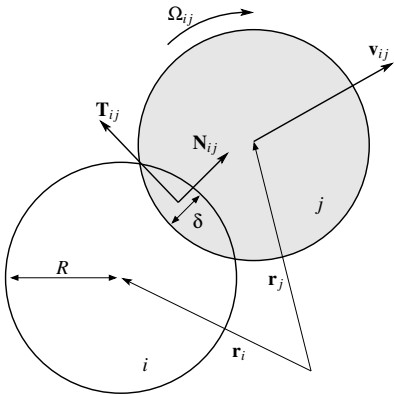

**Figure 2.** Two pebbles in contact, located respectively at $\mathbf{r}_i$ and $\mathbf{r}_j$, with an overlap $\delta = 2R - |\mathbf{r}_i - \mathbf{r}_j|$. $\mathbf{v}_{ij}$ and $\Omega_{ij}$ are, respectively, the translational and angular relative velocities of pebble $j$ with respect to pebble $i$. Normal $\mathbf{N}_{ij}$ and tangential $\mathbf{T}_{ij}$ forces apply at the contact.

| $\Theta - \Theta_c$ | $p$ | $\gamma$ |
|---|---|---|
| 0.015 | 2.11 | 0.21 |
| 0.031 | 2.77 | 0.18 |
| 0.049 | 2.70 | 0.19 |
| 0.071 | 2.00 | 0.24 |

**Table 2.** Coefficients used in the empirical fit of the flux of energy as a function of sediment supply (see equation (27)).





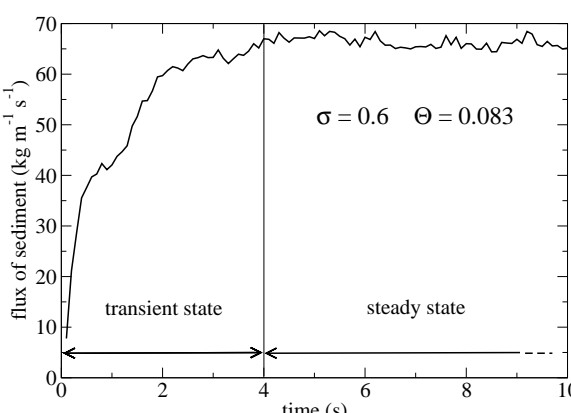

**Figure 3.** Flux of sediment $q(t)$ as a function of time, for a Shields number $\Theta = 0.083$ and a sediment supply $\sigma = 0.6$. A transient state is observed for a few seconds before the steady state is reached.

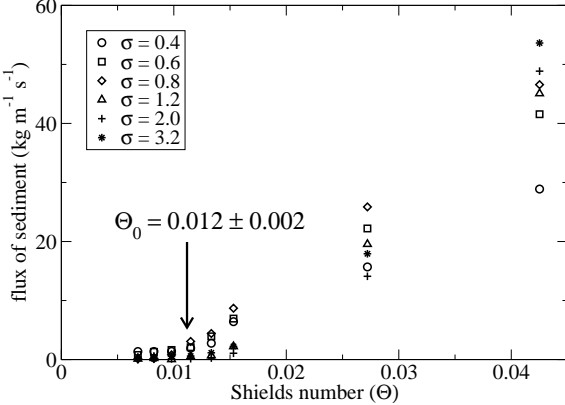

**Figure 4.** Average flux of sediment $Q_s = \langle q_s \rangle$ as a function of the Shields number $\Theta$ and for different values of the sediment supply $\sigma$. $Q_s$ becomes significantly larger than 0 when the Shields number exceeds $\Theta_c \approx 0.012$.




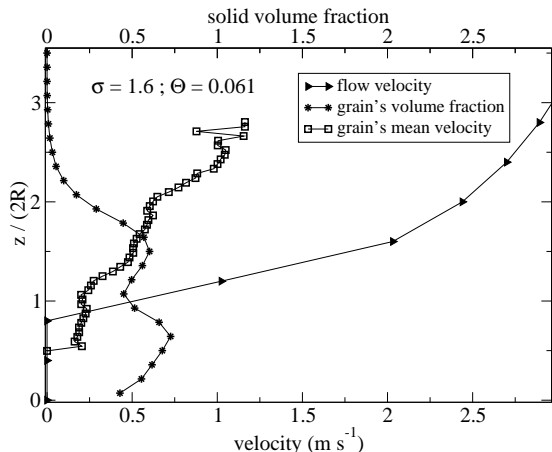

**Figure 5.** Solid volume fraction and velocity of the fluid and the pebbles as a function of height for a Shields number $\Theta = 0.061$ and a sediment supply $\sigma = 1.6$. Two distinct layers of pebbles can be observed. The velocity of the flow and of pebbles vanishes in the bottom layer.

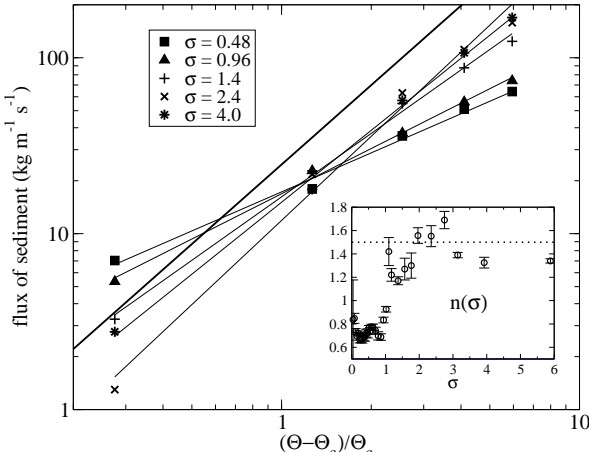

**Figure 6.** Flux of sediment as a function of the relative excess shear stress $(\Theta - \Theta_c)/\Theta_c$ for different values of the sediment supply, in a log-log scale. The critical Shields number is $\Theta_c = 0.012$. Plain lines are best fits by a power law (see equation (22)). The bold curve represents the Meyer-Peter/Müller transport law. *Insert*: exponent of the best fit by a power law, and its standard deviation, as a function of the sediment supply $\sigma$. The dotted line represents $n = 1.5$.



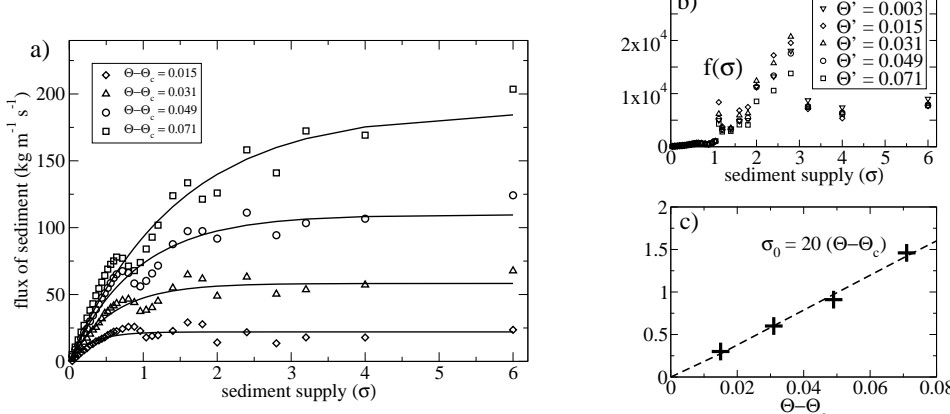

**Figure 7.** a) Flux of sediment as a function of the sediment supply for different values of the Shields number. The flux of sediment increases linearly for low sediment supplies, but tends to a saturated value when transport capacity is reached. Plain lines are best fits by the equation $Q_s(\Theta, \sigma) = Q_t(\Theta) \times (1 - e^{-\sigma/\sigma_0})$. b) The function $f(\sigma)$ (as defined by equation (22)) is shown to be roughly independent of the excess shear stress $\Theta' = \Theta - \Theta_c$. c) Evolution of the critical sediment supply $\sigma_0$ with $\Theta - \Theta_c$. The dashed line is the best linear fit.

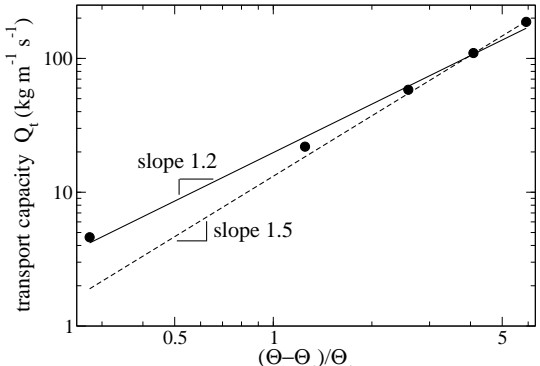

**Figure 8.** Saturated value of the sediment transport rate $Q_t$ as a function of $(\Theta - \Theta_c)/\Theta_c$. The plain line is the best fit by a power law; the dashed line is the fit by a power law of index $n = 1.5$.





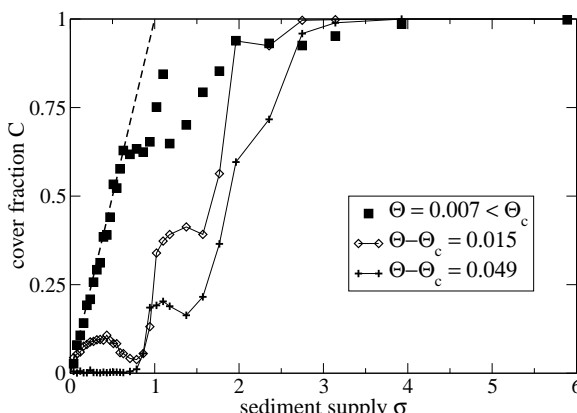

**Figure 9.** Static cover fraction as a function of the sediment supply, for different values of the Shields number. The dashed line represents the function $C = \sigma$.

| $\Theta - \Theta_c$ | $\varphi$ |
|---|---|
| 0.015 | $2.86 \pm 0.03$ |
| 0.031 | $4.65 \pm 0.06$ |
| 0.049 | $6.32 \pm 0.18$ |
| 0.071 | $8.43 \pm 0.26$ |

**Table 3.** Cover factor $\varphi$ extracted from the exponential tail of the cover function $F$ (see equation (33)).



Earth **Surface**
**Dynamics**
Discussions



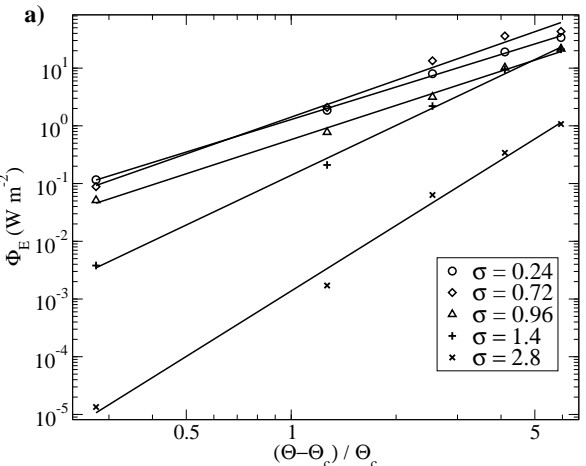

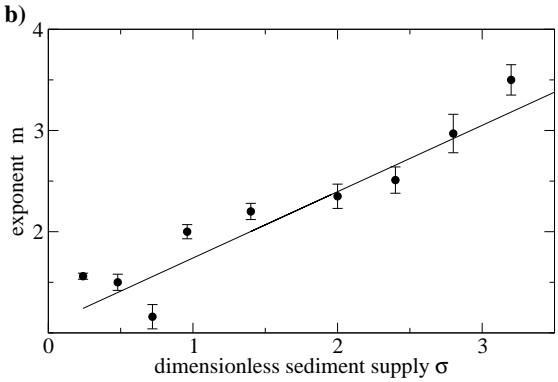

**Figure 10.** a) Flux of energy received by the bedrock as a function of the relative excess shear stress, in a log-log scale. Plain lines are best fits of $\Phi_E$ with power laws (see equation (25)). b) Exponent $m(\sigma)$ of the power law as a function of the dimensionless sediment supply $\sigma$. Data points are fitted by a linear function.





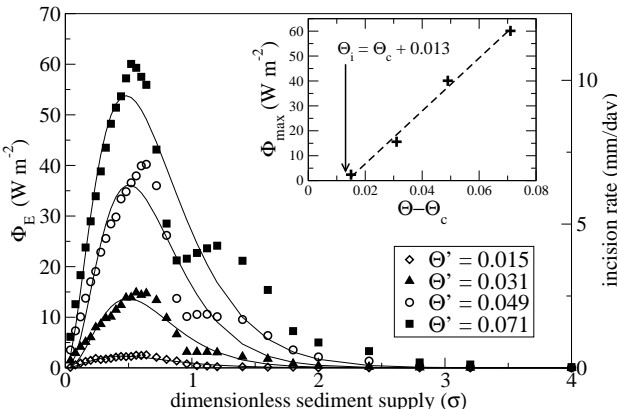

**Figure 11.** Flux of energy delivered to the bedrock as a function of the dimensionless sediment supply. The energy transfer increases when the sediment supply increases in the range $0 < \sigma < 0.5$ ('tool-effect'). It then decays for larger sediment supplies ('cover effect') and vanishes when the sediment supply reaches $\sigma \approx 3$. The maximum flux is reached for a critical value $\sigma_m$ of the sediment supply, that depends only slightly on the Shields number. Plain lines are the best empirical fits by equation (27). The equivalent incision rate (right-side axis) is computed using equation (34) (see section 4.4). The insert plots the maximum value of the energy flux as a function of $\Theta - \Theta_c$. The dotted line is the best affine fine, which reveals the existence of an incision threshold $\Theta_i > \Theta_c$.

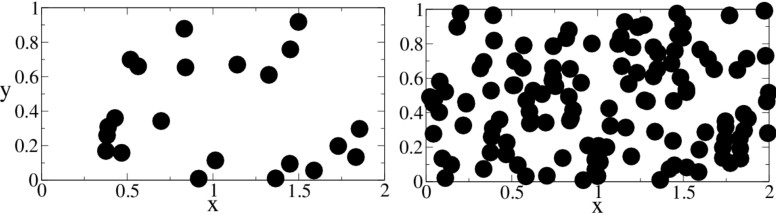

**Figure 12.** Two different cases of bedrock roughness: the positions of the glued spheres are plotted as seen from above. *Left*: $\chi = 0.08$. *Right*: $\chi = 0.52$. The flow is in the $x$-direction, from left to right.





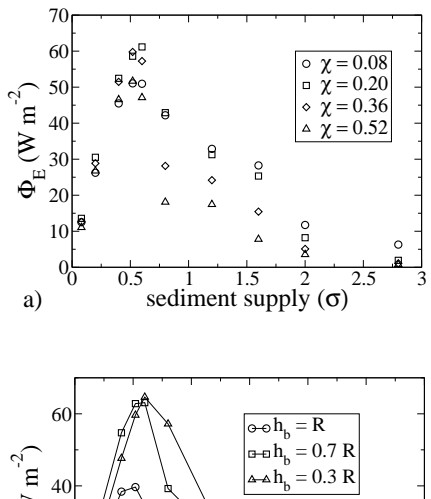

**Figure 13.** Flux of energy delivered to the bedrock as a function of the sediment supply for $\Theta - \Theta_c = 0.071$:
a) for increasing values of the roughness density $\chi$; b) for different values of the roughness height.





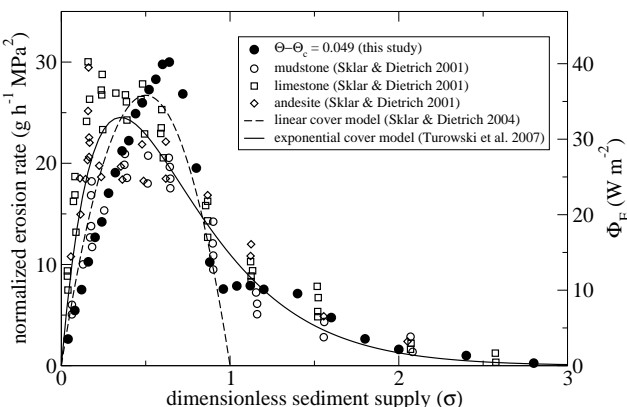

**Figure 14.** We plot on the same graph our numerical prediction for the flux of energy delivered to the bedrock (full circles), erosion rates measured experimentally by Sklar and Dietrich (2001) (empty symbols), and the best fit for these experimental values by the linear cover model of Sklar and Dietrich (2004) and the exponential cover model of Turowski et al. (2007) (plain line). The sediment mass used in the experiments is converted into a dimensionless sediment supply using equation (26). The scale for $\Phi_E$ has been chosen so that its maximum coincides with the experimental maximum erosion rate.





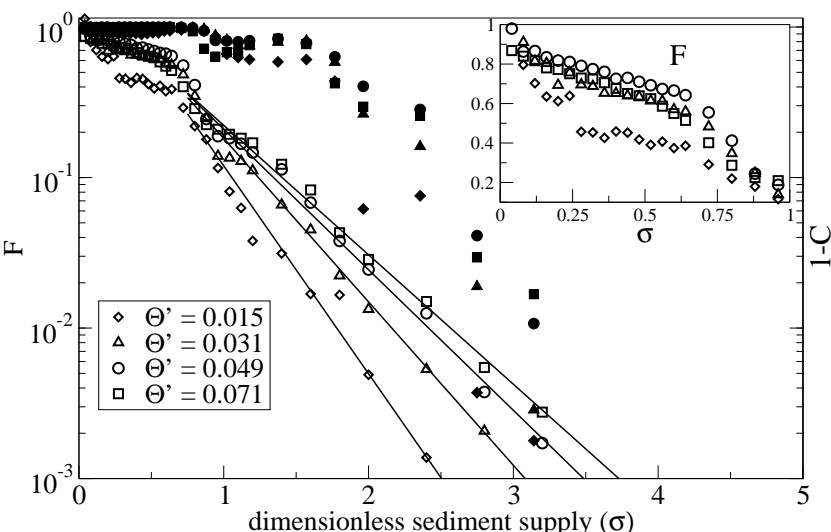

**Figure 15.** Empty symbols: cover function $F$, as defined by equation (32), plotted as a function of $\sigma$, in a log-linear scale. Plain lines are best fits by exponential tails for $\sigma > 1$. Full symbols plot $1 - C$, with $C$ the static cover fraction as computed in section 3.2. *Insert:* Zoom on the cover function for low values of $\sigma$, in a linear scale.