# Peer review of "Bedrock incision by bedload: insights from direct numerical simulations"

_Earth Surface Dynamics, 2015_

## Referee Comment (RC1) · Anonymous Referee #1 · 12 Feb 2016

General comments: The paper reports on numerical simulations of bedload transport and its interaction with the underlying static sediment layer. The model used is based on the discrete element method, which allow for the explicit description of the "sediments" dynamics and trajectories. The interaction with the driving stream is taken into account through effective drag and entrainment. This numerical setting thus gives access to quantitative direct measurements of the corelation between beadload and erosive power. The paper is well written and well organised. The results are original and suggest interesting perspectives using similar approaches. I have however few comments that I raise in the following, and leave it to the authors to decide whether or not they want to add a line on these in their manuscript.

Specific comments:
[Figure]

p4: I understand why using spherical grains makes the computation much easier. However it is clear that the effect of the shape is non trivial and non negligeable. Hence my question: is there a typical shape of grains at the surface of river bed? Could it be that beyond dynamical consideration, saltating grains and covering grains belong to a different class in terms of shape/ aspect ratio etc...? do we have any insight in these, including experimental data on the efficiency of tranport (namely lift and drag) for different class of grains?

p5, line 119: it seems very unlikely that comminution would affect the pebble over such short time scale as you consider indeed, the comment seems unnecessary (or maybe prompted by earlier referee?)

p7, line 190-195. Why the value phi_B=0.5 specifically? Why this simplification of the velocity profile, since this aspect does not seem to be CPU expensive (unless I am mistaken on this last point?)

p8, line 215-220. It seems that the general behaviour is dominated by binary collision. Can we imagine that trajectories between two collisions may be predicted analytically and hence the time step be adaptative, as in event-driven methods? That could permit to greatly increase the duration of the experiment at little cost.

Very general question: I have the idea (could be partly wrong) that mechanical incision occurs during rare catastrophic events during which larger than usual pebbles will be transported by a highly energetic flow, and just plough the river bed. That is, mechanical incision may no be the result of a regular process as you describe. How much the regime you simulate is "catastrophic" compared to normal flow condition?

- References: I am a bit surprised to find no Métivier et al on experimental beadload and no Ancey et al (particularly with T. Bohm or J. Heyman) who has addressed similar issues in recent years. This should certainly be added (= the minor revision of the recommendation).

**ESurfD**

Interactive
comment

---

## Referee Comment (RC2) · Anonymous Referee #2 · 3 Mar 2016

This paper investigates the theoretical bedrock incision model of Sklar and Dietrich (2001) using discrete element modeling to determine the functional dependence the Shield's number has on incision rates, as well as the dependence of sediment supply on the "cover-effect". The results of this research show similar bedrock incision trends to those expected from theory and recorded by experiments and also "predict[s] that the cover term should decay linearly at low sediment supply and exponentially at high". This work provides new insight into the mechanics of bedload transport and bedrock incision and is therefore worthy of publication. The introduction and methods are both well written, however, the results and discussion have parts that are a bit difficult to follow. Although I do not have any major reservations about the paper, I have a few suggestions and comments that I will list below in order as they appear in the manuscript.

[Figure]

15-84: The introduction is well written and provides a good background on the mechanics of berock erosion.

87-91: I would like to see more discussion of the use of spherical particles, since angularity has been shown to change the transport. If it would be easy to do, it would be interesting to see how the results would change for angular grains or simply cite the work on transport of angular grains and speculate how this simplification of spherical grains can effect to outcomes.

118: You use a single grain size population for the simulations, however grain hiding and protrusion due to relative grain size differences with neighboring pebbles has been shown to effect bedload transport and could therefore have an effect on incision rates.

153-155: How do these values for collision duration and coefficient of restitution compare with real values for bedload impacts? Schmeeckle et al (2001) provides good experimental results indicating coefficient of restitution of 0.65 for collision stokes numbers over 105.

240-363: The results provide a lot of information to the reader and can be a little overwhelming especially with the many figures presented, where some results are only presented for certain values of supply (line 261). It might be clearer to the reader if you present a table of all simulations with corresponding input parameters, then you could refer to those simulations when referring to which data is used for certain plots.

255: You compare your results to other "experimental observations" so a citation is needed.

257: Reference "most observations of the transport threshold" so citation is needed.

276: guarantee spelled wrong.

305: "This finding is once again consistent with most transport models" needs citation.

365-542: The discussion has organized well but just had a few areas that need clarification.

367: "appears to be consistent with experimental observations" needs citation.

379-381: Sentence starting "Let us note. . ." is worded weirdly and should be rewritten for clarity.

385: Sentence starting with "Once rescaled with the value of threshold. . .", please note which value you used.

386: "consistent with common measurements" needs citation.

422-432: Does the model take into account abrasion that can occur due to frictional sliding between bedload and bedrock? Gabor and Domokos (2012) and Litwin Miller et al. (2014) show that frictional abrasion, in addition to collisional abrasion played a role in overall pebble abrasion. This consideration could account for why you do not see any incision while the pebbles are only rolling while the experiments of Sklar and Deitrich (2001) did.

539: How do your simulation input parameters, as well as rock material properties (tensile strength, etc.) compare with the site in Taiwan?

539-542: If it is possible, I think that running a simulation to determine the long-term average incision rates would make the conclusions of this research much stronger.

576: "Though our results are qualitatively consistent with experimental observations and another type of models. . ." needs citation.

Figure 4: Label axis as "average flux" to differentiate from flux in figure 3.

Figure 7: Label y-axes for b) and c).

[Figure]

---

## Author Comment (AC1) · 21 Mar 2016

**Reply to the comments of referees 1 and 2 on the manuscript esurf-2015-56**
*Bedrock incision by bedload: insights from direct numerical simulations,*
by G. Aubert, V.J. Langlois and P. Allemand.

We would like to thank the two anonymous referees for their careful reading of our article and their insightful comments. We detail below the modifications we have brought to the article, according to their suggestions.

**Anonymous Referee # 1**

*p4: I understand why using spherical grains makes the computation much easier. How- ever it is clear that the effect of the shape is non trivial and non negligeable. Hence my question: is there a typical shape of grains at the surface of river bed? Could it be that beyond dynamical consideration, saltating grains and covering grains belong to a different class in terms of shape/ aspect ratio etc...? do we have any insight in these, including experimental data on the efficiency of tranport (namely lift and drag) for different class of grains?*

We added a few lines of discussion about the possible influence of irregular/angular shapes at the end of section 5 and refer to the works of Rust [1972] and Komar and Li [1986]. Though it is possible to model such objects by assembling (with elastic beams) several spherical particles into a single pebble of irregular shape, we do not believe that our simple fluid model would be very appropriate to compute their dynamics within a turbulent flow.

*p5, line 119: it seems very unlikely that comminution would affect the pebble over such short time scale as you consider indeed, the comment seems unnecessary (or maybe prompted by earlier referee?)*

This precision was indeed added following an earlier review. It might seem obvious that comminution does not affect pebbles over a duration of one minute but at this stage of the article we have not yet precised on which timescale our simulations operate and we prefer to keep it clear from the beginning that we do not see explicitly in our model long-term abrasion and comminution, but only infer them from the energy lost during impacts.

*p7, line 190-195. Why the value $\phi_B = 0.5$ specifically? Why this simplification of the velocity profile, since this aspect does not seem to be CPU expensive (unless I am mistaken on this last point?)*

A more accurate description of the influence of the particles on the flow field would indeed be beneficial, but it would in fact be somewhat CPU expensive, considering that the horizontal momentum transfered from the fluid flow to the pebbles would have to be computed at each timestep (or every few timesteps), and if we adopt the same timestep as for the granular dynamics, the velocity profile then tends to diverge. As shown by Duran et al. [2012] with such a model (for thick sediment layers), the velocity profile is essentially zero within the layer and equal to the turbulent logarithmic profile when $\phi \gtrsim 0.5$. There is a very thin transition zone (of the order of one grain diameter), which we neglect in our model.

Besides, as we discuss in the final paragraph, computing a horizontally-averaged velocity profile in our case (with low sediment loads) is not likely to be much more accurate than our estimate: for $\sigma \lesssim 2$, the pebbles are not evenly distributed on the bedrock, and we should expect many lateral and longitudinal heterogeneities in the velocity field.

*p8, line 215-220. It seems that the general behaviour is dominated by binary collision. Can we imagine that trajectories between two collisions may be predicted analytically and hence the time step be adaptative, as in event-driven methods? That could permit to greatly increase the duration of the experiment at little cost.*

An event-driven method would be relevant if collisions were only binary and non-persistent. This is not always the case in our simulations: on one hand, most of the collisions between a pebble

and the bedrock are indeed binary, but some pebbles are rolling along the bedrock, which means a permanent contact. Some also get stuck within the roughness with the same consequence. On the other hand, most collisions between mobiles pebbles are very quick but can frequently (when the sediment load is large enough) involve more than 2 particles. The event-driven method would be favourable only in the case of a very low sediment load, in which case the computational cost is not too high even with our integration method.

*Very general question: I have the idea (could be partly wrong) that mechanical incision occurs during rare catastrophic events during which larger than usual pebbles will be transported by a highly energetic flow, and just plough the river bed. That is, mechani- cal incision may no be the result of a regular process as you describe. How much the regime you simulate is catastrophic compared to normal flow condition?*

The prevalence of large catastrophic hydraulic events on long-term incision has indeed been suggested by some studies. However, let us note that these events can also provide the river with a very large sediment load, which could result in a total inhibition of abrasion by cover, as suggested, for instance, by the numerical model of Lague [2010].

Though our model does not take into account incision by plucking, let us though insist on the fact that the energy delivered to the bedrock increases faster than linearly with the Shields number (see our figure 10b). This tends to increase the influence of large hydraulic events, assuming that the cover fraction is not too large. We added a few lines to discuss this in section 4.2.

*References: I am a bit surprised to find no Metivier et al on experimental beadload and no Ancey et al (particularly with T. Bohm or J. Heyman) who has addressed similar issues in recent years. This should certainly be added (= the minor revision of the recommendation).*

We added several references to some of the suggested papers (with in particular an additional discussion of the observed fluctuations in section 3.1), plus a reference to Lajeunesse et al. [2010].

**Anonymous Referee # 2**

*87-91: I would like to see more discussion of the use of spherical particles, since angularity has been shown to change the transport. If it would be easy to do, it would be interesting to see how the results would change for angular grains or simply cite the work on transport of angular grains and speculate how this simplification of spherical grains can effect to outcomes.*

We added the following lines to the discussion: The assumption of spherical particles is also likely to have an effect on our predictions: indeed, angular or irregularly shaped sediments would probably be less easily put into motion by the flow (for instance, a flat pebble would be harder to dislodge from the bedrock [Rust, 1972; Komar and Li, 1986]). In contrast, once entrained by the flow they could slide along the bedrock (instead of simply rolling) and therefore contribute to its wear via solid friction.

Within the same numerical frame, it would be possible to model pebbles of more complex shapes by 'glueing' together (via elastic beams) a few spherical particles. However, the hydrodynamical properties of such particles would be less easy to infer and our simple fluid model would probably be too simplistic in this case.

*118: You use a single grain size population for the simulations, however grain hiding and protrusion due to relative grain size differences with neighboring pebbles has been shown to effect bedload transport and could therefore have an effect on incision rates.*

The effect of the possibly wide size distribution within the bedload layer is indeed likely to be of importance, since we expect various processes like size-sorting and armouring to happen. A relevant simulation of these long-term effects, however, would probably require to work at a larger lengthscale and a longer timescale, which we should be able to do in the foreseeable future by

running our code on GPUs.

*153-155: How do these values for collision duration and coefficient of restitution com- pare with real values for bedload impacts? Schmeeckle et al (2001) provides good experimental results indicating coefficient of restitution of 0.65 for collision stokes num- bers over 105.*

The data obtained by Schmeeckle et al. [2001] for natural sediment show that in many cases (even at high Stokes number), the coefficient of restitution is below the 0.65 value predicted by Davis. This prediction, however, is valid for elastic spheres. In our simulation, since we want to simulate abrasion of the bedrock, we have to take into account the fact that during an impact, our particles lose some energy due to mechanical damage (wear, small cracks, etc). This is why we chose a value of $e$ within the lower range of the experimental measurements. We added a discussion about this in section 2.2.

*240-363: The results provide a lot of information to the reader and can be a little overwhelming especially with the many figures presented, where some results are only presented for certain values of supply (line 261). It might be clearer to the reader if you present a table of all simulations with corresponding input parameters, then you could refer to those simulations when referring to which data is used for certain plots.*

We understand that there are many results presented in this section; however they correspond to simulations run with only two control parameters ($\sigma$ and $\Theta$). Since we have performed simulations for 5 values of the Shields number and 32 values of the sediment supply, we are not convinced that referring to a table of labelled runs, rather than directly to the values of $\sigma$ and $\Theta$, would really be of help to the reader.

*255: You compare your results to other experimental observations so a citation is needed. 257: Reference most observations of the transport threshold so citation is needed.*

We added the reference to the review by Lamb et al. [2008], which we already used in the more detailed discussion of this point in section 4.1.

*276: guarantee spelled wrong*

Corrected.

*305: This finding is once again consistent with most transport models needs citation*

The number of papers dealing with the bedload transport law being considerable, we only added a reference to Lajeunesse et al. [2010] who review a number of them.

*365-542: The discussion has organized well but just had a few areas that need clarification. 367: appears to be consistent with experimental observations needs citation*

As explained in the following lines, we here refer to figure 6, where we compare the order of magnitude of our predicted flux of sediment with the common empirical Meyer-Peter/Müller law.

*379-381: Sentence starting "Let us note..." is worded weirdly and should be rewritten for clarity*

We agree with the referee that this sentence was awkward to follow and reformulated it as follows: *Let us note that Duran et al. [2012], who also used spheres but modeled more sophisticated fluid-grain interactions, obtained a critical Shields number of* 0.12.

*385: Sentence starting with "Once rescaled with the value of threshold": please note which value you used.*

We added the value $\Theta_c = 0.012$.

*386: consistent with common measurements needs citation*

We added here a reference to the Meyer/Peter-Müller article, whose empirical fit appears in figure 6.

*422-432: Does the model take into account abrasion that can occur due to frictional sliding between bedload and bedrock? Gabor and Domokos (2012) and Litwin Miller et al. (2014) show that frictional abrasion, in addition to collisional abrasion played a role in overall pebble abrasion. This consideration could account for why you do not see any incision while the pebbles are only rolling while the experiments of Sklar and Deitrich (2001) did.*

The frictional dissipation of energy can be taken into account in our simulations through the dissipative work of the (tangential) contact forces. However, our particles being spherical, they always tend to roll without sliding on the bedrock, therefore not experiencing much solid friction along the bedrock. The angularity of the pebbles would certainly increase the frictional component of abrasion. This would be achievable with our simulations, without losing the advantages of spherical particles, by considering the motion of pebbles made of a rigid cluster of spheres, but for which our simple hydrodynamical model would probably not be appropriate.

*539: How do your simulation input parameters, as well as rock material properties (tensile strength, etc.) compare with the site in Taiwan?*

Considering the simplicity of our numerical model, we prefer to compare only the order of magnitude of our predicted short-term incision rate, which validates our approach. A more detailed comparison the would probably not be relevant.

*539-542: If it is possible, I think that running a simulation to determine the long-term average incision rates would make the conclusions of this research much stronger.*

This is indeed an interesting potential outcome of our study but this would go beyond the scope of the present article. Indeed our numerical prediction is very general, whereas an actual simulation of the long-term incision rate would require to implement this predicted incision rate (as a function of both sediment load and discharge) within a river evolution model (such as, for instance, SSTRIM [Lague, 2010]), which we suggest in the conclusions. In such a study the precipitation and sediment influx distribution functions would be highly dependent on the chosen river. We are presently working on first adapting our model in order to compute the incision rate of the river banks, which is also critical in the long-term evolution of a river bed: even when the bedrock is fully protected by the bedload layer, the banks can still be exposed to lateral impacts, which would result in the river bed widening. This effect has so far been modeled without any strong experimental nor numerical constraints. Obtaining numerical predictions for this abrasion of banks as well as for the comminution of the pebbles themselves, coupled with the present prediction for bedrock abrasion, would allow to run realistic simulations of the long-term evolution of a river bed.

*576: "Though our results are qualitatively consistent with experimental observations and another type of models..." needs citation*

We added references to Sklar and Dietrich [2001] for the experiments and Sklar and Dietrich [2004] and Turowski et al. [2007] for the numerical approaches.

*Figure 4: Label axis as average flux to differentiate from flux in figure 3. Figure 7: Label y-axes for b) and c).*

These legends have been added to the corresponding figures.

**References**

Orencio Duran, Bruno Andreotti, and Philippe Claudin. Numerical simulation of turbulent sediment transport, from bed load to saltation. *Physics of Fluids*, 24(10):103306, 2012. ISSN 10706631. doi: 10.1063/1.4757662.

Paul D Komar and Zhenlin Li. Pivoting analyses of the selective entrainment of sediments by shape and size with application to gravel threshold. *Sedimentology*, 33(3):425–436, 1986.

D. Lague. Reduction of long-term bedrock incision efficiency by short-term alluvial cover intermittency. *Journal of Geophysical Research*, 115(2):F02011, May 2010. ISSN 0148-0227. doi: 10.1029/2008JF001210.

E. Lajeunesse, L. Malverti, and F. Charru. Bed load transport in turbulent flow at the grain scale: Experiments and modeling. *Journal of Geophysical Research*, 115 (F4):F04001, October 2010. ISSN 0148-0227. doi: 10.1029/2009JF001628. URL http://doi.wiley.com/10.1029/2009JF001628.

Michael P Lamb, William E Dietrich, and Jeremy G Venditti. Is the critical shields stress for incipient sediment motion dependent on channel-bed slope? *Journal of Geophysical Research: Earth Surface*, 113(F2), 2008.

Brian R Rust. Pebble orientation in fluvial sediments. *Journal of Sedimentary Research*, 42(2), 1972.

Mark W Schmeeckle, Jonathan M Nelson, John Pitlick, and James P Bennett. Interparticle collision of natural sediment grains in water. *Water Resources Research*, 37(9):2377–2391, 2001.

Leonard S. Sklar and William E. Dietrich. Sediment and rock strength controls on river incision into bedrock. *Geology*, 29(12):1087, 2001. ISSN 0091-7613.

Leonard S Sklar and William E Dietrich. A mechanistic model for river incision into bedrock by saltating bed load. *Water Resources Research*, 40(6), 2004.

Jens M. Turowski, Dimitri Lague, and Niels Hovius. Cover effect in bedrock abrasion: A new derivation and its implications for the modeling of bedrock channel morphology. *Journal of Geophysical Research*, 112(F4):F04006, November 2007. ISSN 0148-0227. doi: 10.1029/2006JF000697.